# Loss of Human Epidermal Receptor 2 Expression in Formalin-Fixed Paraffin-Embedded Breast Cancer Samples and the Rescuing Effect of Enhanced Antigen Retrieval and Signal Amplification

**DOI:** 10.3390/life14010031

**Published:** 2023-12-25

**Authors:** Xiuli Ma, Lixin Zhou, Qi Wu, Ling Jia, Xinting Diao, Qiang Kang, Xiaozheng Huang, Yiqiang Liu, Taobo Hu, Mengping Long

**Affiliations:** 1Department of Pathology, Peking University Cancer Hospital, Beijing 100083, China; maxiuli10@sina.com (X.M.); zlixin@aliyun.com (L.Z.); wuqi0812@163.com (Q.W.); nannan813@163.com (L.J.); 15116951640@163.com (X.D.); wanwan927@sohu.com (Q.K.); hxz182182@163.com (X.H.); victor.liu76@163.com (Y.L.); 2Department of Breast Surgery, Peking University People’s Hospital, Beijing 100044, China

**Keywords:** breast cancer, HER2-low, antigenicity loss, immunohistochemical stain

## Abstract

As an important therapeutic target in breast cancer, HER2 expression assessed by immunohistochemistry plays a critical role in breast cancer treatment. Recent advances in HER2 antibody–drug conjugate therapy have enabled patients with HER2-low expression breast cancer to benefit from the drugs. However, it is not known whether the HER2-low expression in breast cancer FFPE blocks would be lost as storage time increased. In this study, we aimed to assess the loss of HER2 antigenicity in stored FFPE blocks of breast cancer and the rescue effect of modifying the protocol of antigen staining. We selected archived HER2-low breast cancer FFPE blocks with stored time ranging from 1 year to over 15 years and re-detected the expression of HER2. Our study showed that HER2 antigenicity loss increased with storage time and could cause false negativity in HER2-low detection. Moreover, we showed that by either increasing the antigen retrieval time or applying the tyramide signal amplification (TSA) kit, the HER2 signal can be rescued and detected in about half of the cases with HER2-low loss without causing false positivity.

## 1. Introduction

Breast cancer is the most prevalent malignancy in women [1]. Current therapy for breast cancer is heavily dependent on biomarker detection and evaluation in breast cancer tissue [2]. HER2 (human epidermal receptor 2) is a critical biomarker for the subtyping and targeted therapy of breast cancer [3,4]. The expression of HER2 is routinely assessed by IHC (immunohistochemistry) staining in FFPE (formalin-fixed paraffin-embedded) tissue. The IHC staining signal is then interpreted in a semiquantitative way by categorizing the signal into a four-grade system ranging from 0 to 3+ [5]. Breast cancer that is HER2 3+ by IHC or HER2 2+ by IHC with ISH (in situ hybridization) amplification is defined as HER2-amplified breast cancer, while others are defined as HER2 non-amplified. Patients with HER2-amplified breast cancer can benefit from anti-HER2 antibodies, including trastuzumab and pertuzumab [5]. Recent studies found that among patients with HER2 non-amplified breast cancer, those with HER2 1+ or 2+ by IHC can also benefit from anti-HER2 ADCs (antibody-drug conjugates including ado-trastuzumab emtansine and fam-trastuzumab deruxtecan-nxki), while those without HER2 expression (HER2-zero breast cancer) cannot [6,7,8,9]. Breast cancer with HER2 expression but not amplification is called HER2-low breast cancer [8,10]. HER2-low breast cancer accounts for 60–80% of all breast cancer and is currently receiving intensive attention in the field of breast cancer [9,11]. Thus, the accurate detection of HER2 expression and the distinguishing between HER2-zero and HER2-low breast cancers have become important for the treatment and research in the breast cancer field [12].

As FFPE samples are routinely archived in the tissue bank and the investigation of a prognostic and predictive biomarker is often retrospective, the detection of HER2 expression in aged FFPE breast cancer tissue is frequently needed. Previous studies have reported the loss of staining intensity or antigenicity of HER2 in stored breast cancer tissues, and the loss of antigenicity is positively correlated with the storage time [13,14,15,16]. Thus, it is possible that the identification of HER2-low breast cancer can be false-negative and be classified as HER2-zero in a stored breast cancer tissue block due to the loss of HER2 antigenicity. However, the proportion of false negativity due to time change is currently unknown. The detection of protein expression by IHC staining is based on the specific binding of the applied antibody to the targeting protein in tissue. Besides the antibody used for detection, the final signal can also be affected by the degree of antigen unmasking by antigen retrieval as well as the visualization method [17,18]. Thus, it is possible that the loss of HER2 antigenicity in stored FFPE blocks can be rescued by modification according to the IHC procedure, including increasing the antigen retrieval time and applying tyramide signal amplification (TSA). In this study, we aimed to assess the loss of HER2 antigenicity in stored FFPE blocks of breast cancer and the rescue effect of modifying the protocol of antigen staining.

## 2. Methods

### 2.1. Tissue Selection

All of the FFPE blocks were selected from the archived biobank of the pathological department at Peking University Cancer Hospital (from 2002 to 2022). The diagnosis information and the corresponding tissue were retrieved from the electronic pathological system. All of the blocks were stored at room temperature in the same condition. A total of 96 breast cancer FFPE blocks were selected from the archived biobank in the pathology department at Peking University Cancer Hospital. They were divided into five groups according to the stored age, including <1 year, 2–4 years, 5–10 years, 11–15 years and >15 years. Since the number of FFPE blocks stored over 15 years was limited, we were only able to find 5 cases in group “>15 years”, while the case numbers in other groups ranging from G1 to G4 were 26, 28, 17 and 20. The original HER2 IHC slide was reviewed and scored by two pathologists to define the “original HER2 score” of the tissue. In cases where the original slide was not available, the “original HER2 score” was derived from the pathology report.

### 2.2. Immunohistochemical Staining

The FFPE block was cut into a 4 µm slide. The immunohistochemistry staining of HER2 was performed on the automated Bench Mark Ultrasystem using the VENTANA 4B5 antibody (Roche, Tucso, AZ, USA), following the manufacturers’ instructions. The specific condition for antigen retrieval was a pH8.4 EDTA solution incubated at 95 °C for 36 min. For the original HER2 stain, cases from the year 2008 and later were stained using the VENTANA 4B5 antibody, while cases before 2008 were stained with the HER2 antibody of clone CB11 (Abcam, Boston, MA, USA).

### 2.3. Pathology Evaluation

HER2-low status was defined as IHC1+ or IHC 2+/in situ hybridization negative, and HER2-zero was defined as IHC0+, based on the American Society of Clinical Oncology/College of American Pathologists guidelines. A cutoff of >10% cell staining for HER2-positivity was applied. The evaluation was performed by two board-certified pathologists (M.L. and Y.L.).

### 2.4. Data Analysis

For comparison of datasets, a chi-square test (X^2^) was used, with *p* < 0.05 considered statistically significant.

### 2.5. Tissue Antigen Retrieval (TAR) Elongating and TSA Signal Amplification Applying

TAR elongating was performed by increasing the antigen retrieval time from 36 min to 92 min, with other conditions remaining unchanged. For conventional IHC detection, the ultraView Universal DAB Detection Kit (Cat. 760–500, Roche, Tucso, AZ, USA) was used after the incubation of the primary anti-HER2 antibody. For TSA, the OptiView DAB IHC Detection Kit (Cat. 760–700, Roche, Tucso, AZ, USA) together with the OptiView Amplification Kit (Cat. 760–099, Roche, Tucso, AZ, USA) were used according to the manufacturer’s specifications. All of the staining processes were performed using the Ventana benchmark ultra system. The study design is shown in Figure 1.

## 3. Results

### 3.1. Loss of HER2 Antigenicity in FFPE Blocks of Breast Cancer with Stored Age

Among the 96 breast cancer blocks selected in our study, the number of HER2 0, HER2 1+, HER2 2+ and HER2 3+ cases were 6(6/96, 6.3%), 69(69/96, 71.9%), 15(15/96,15.6%) and 6(6/96,6.3%), respectively. All of the HER2 2+ cases were verified to be HER2 non-amplified using fluorescent in situ hybridization (FISH). Thus, the HER2-low cases accounted for 87.5% of the selected blocks, while the HER2 1+ cases accounted for 82.1% of the HER2-low population. All of the blocks were sectioned and stained with HER2-antibody (clone4B5) using the Ventanna system. The stained slides were reviewed and scored by two pathologists. We first analyzed the effect of stored time on HER2 loss in all of the HER2-expressed blocks, including HER2 1+, HER2 2+ and HER2 3+ (Figure 2). We defined the loss of one HER2 intensity score, including 3+ to 2+, 2+ to 1+ and 1+ to 0 as LOH1, and the loss of two intensity scores, including 3+ to 1+ and 2+ to 0 as LOH2. The percentage of cases with HER2 intensity loss was 7.7%, 57.1%, 52.9%, 65% and 40%, respectively, from the <1-year group to the >15-year group. It was found that compared with the <1-year group, HER2-expressed blocks stored for 2–4 years, 5–10 years, or 11–15 years all have significantly higher HER2 intensity loss (*p* = 0.000129, 0.001948, 0.000112, 0.0318) (Figure 2).

While among the three latter groups, no significant difference was found in terms of HER2 intensity loss by IHC. It suggests that HER2 antigenicity loss increases with stored time but tends to stabilize after 1 year over a period of up to 15 years. Next, we looked at the percentage of cases that changed from HER2-low to HER2-zero (HER2-low loss) in different time groups (Figure 3). The percentage of cases with HER2-low loss from the <1-year group to the >15-year group was 9%, 58%, 47%, 63% and 100%, respectively. Again, all of the three time groups stored for >1 year have a significantly higher percentage of HER2-low loss than that of the <1-year group. Moreover, no significant difference was detected among the three longer-time groups. The transition from HER2-low to HER2-zero can be contributed by HER2 2+ to HER2 0 as well as by HER2 1+ to HER2 0; the latter is intuitively more likely to occur. Thus, we further analyzed the percentage of cases that changed from HER2 1+ to HER2 0 in each time group. The percentage of cases tchanged from HER2 1+ to HER2 0 in the five time groups was 10%, 65%, 55%, 73% and 100%, respectively, from the <1-year group to the >15-year group (Figure 4). As expected, the ratio of HER2 loss in HER2 1+ cases was higher than that in HER2-expressed cases. Moreover, it showed the same trend as in HER2-expressed cases, in which the loss tends to be stable after 1 year of storage. The above results suggested HER2 antigenicity loss, including HER2 intensity loss and HER2-low loss, both increase with stored time, and the loss becomes stable after 1 year of storage over a period of up to 15 years.

### 3.2. The Rescuing Effect of Modified Immunohistochemical Staining

Since our results suggested that HER2 expression was lost during the storage of FFPE blocks, we further explored whether the modification in the process of IHC staining could help rescue that loss. Two methods were tried, including enlongating the time of TAR and applying tyramide signal amplification (TSA). Antigen retrieval is the process of unmasking the protein crosslinked by fixation with formalin [19,20]. In standard HER2 IHC staining, antigen retrieval is performed by heating the slide at 95 °C for 36 min. Since the degree of antigen unmasking can be affected by the heating time, we extended the heating time to 92 min to see the effect of antigen rescue in HER2 loss cases. Moreover, since the sensitivity of the detection system can also affect the final signal, we applied the tyramide signal amplification (TSA) detection system after the incubation of the primary antibody.

A total of 21 cases that changed from HER2 1+ to HER2 0 were selected for the rescuing. The two methods were separately applied in all of the 21 blocks. A control comprising tissue with HER2 0, 1+, 2+ and 3+ was applied to each stained slide. Neither TAR enhancing nor TSA would cause a change in the staining intensity of the controlled tissue (Figure 5). In the 21 cases with HER2 expression loss from HER2 1+, 10 cases were rescued back into HER2 1+, while the other 11 cases were still HER2 0 when TAR enhancing was applied. As for TSA, 9 out of the 21 cases were recovered to HER2 1+, which are all included in the successfully rescued cases in the TAR enhancing method (Figure 6). Our results showed that both methods can rescue about half of the cases with HER2 loss and would not cause false positivity. Thus, in a FFPE block stored for a period of over 1 year, either TAR enhancing or TSA applying can be considered for HER2 expression detection.

## 4. Discussion

Our results suggest that HER2 antigenicity loss increases with stored time and could cause the transition of HER2-low case to HER2-zero. The antigenicity loss tends to be stable after 1 year over a period of up to 15 years. Thus, in clinical use, the block stored over 1 year for detection of HER2-low false negativity should be at attention. Our study also showed that both TAR enhancing and TSA can rescue the HER2-low loss in about half of the cases without causing false positivity. Although the loss of HER2 antigenicity in breast FFPE blocks caused by stored time has been previously reported, our study was the first to show that in the new terms of HER2-low breast cancer, the loss of HER2 expression can cause false-negativity of HER2-low detection in stored tissue. Different from previous studies, we found that the percentage of cases with HER2 signal loss remains relatively unchanged after a period of 1 year. Our study was limited by the number of cases selected. Also, the variation between the interpretations of HER2 0 and HER2 1+ was reported to be high between different observers. Although current clinical practice uses the IHC stain as a predominant method to evaluate the expression of HER2 in breast cancer, it does have limitations. For instance, IHC staining of FFPE blocks requires the tissue to be fixed with formalin, which could irreversibly destroy the three-dimensional and antigenic structure of the protein and affect detection [21,22]. Also, since IHC is a semiquantitative method, more precise detection of HER2 should be based on quantitative strategies like liquid chromatography mass spectrometry (LC–MS) using fresh frozen tissue [23]. Moreover, previous studies have shown that breast tissues defined as HER2 0 by IHC have HER2 receptors of approximately 20,000 per cell [24]. Thus, IHC is not sensitive enough to measure the number of receptors when it has less than 20,000 receptors per cell.

For the rescuing effect of enhancing TAR and applying a signal amplification system, our result showed that about half the HER2 loss from 1+ to 0 can be rescued. Also, neither method will change what was originally HER2-zero into HER2-low expression. Thus, these two methods should be considered for retrospective clinical studies using archived breast cancer FFPE blocks.

## 5. Conclusions

Our study showed that in FFPE blocks of breast cancer, HER2 antigenicity loss increased with storage time and can cause false-negativeHER2-low detection. Moreover, we showed that both TAR enhancing and TSA can rescue the HER2-low loss in about half of the cases without causing false positivity.

## Figures and Tables

**Figure 1 life-14-00031-f001:**
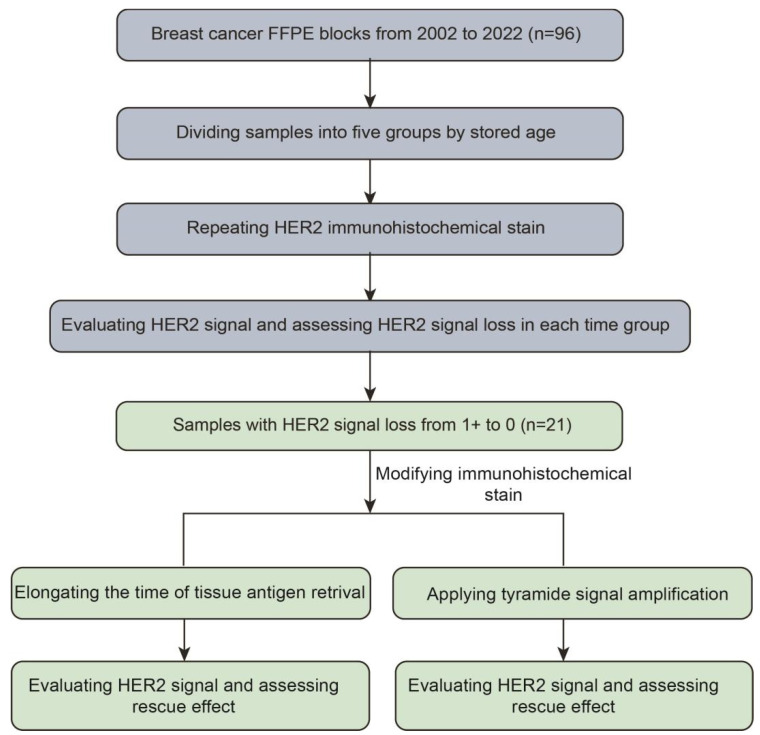
Flow chart of study design.

**Figure 2 life-14-00031-f002:**
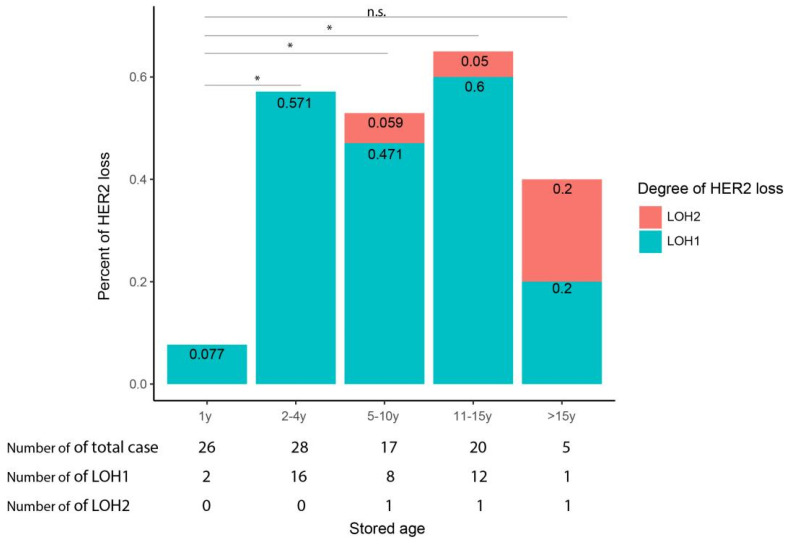
Loss of HER2 staining signal in breast cancer tissue with different stored ages. The selected breast cancer blocks are divided into five groups according to their stored age, and the loss of HER2 staining intensity is summarized. Loss of one HER2 intensity score is designated as LOH1, and loss of two intensity scores is designated as LOH2. * means *p* < 0.05; n.s. means not significant.

**Figure 3 life-14-00031-f003:**
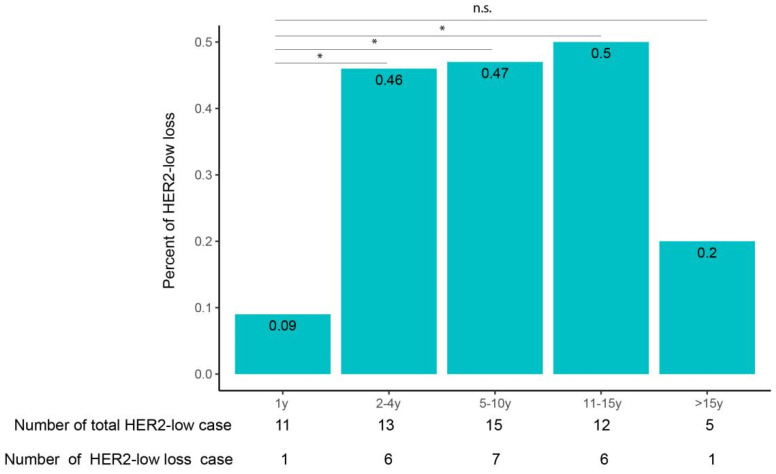
Loss of HER2-low intensity in breast cancer tissue with different stored ages. HER2-low breast cancer stored for different ages was selected and restained with HER2. The proportion of cases with complete loss of HER2 intensity was summarized. * means *p* < 0.05; n.s. means not significant.

**Figure 4 life-14-00031-f004:**
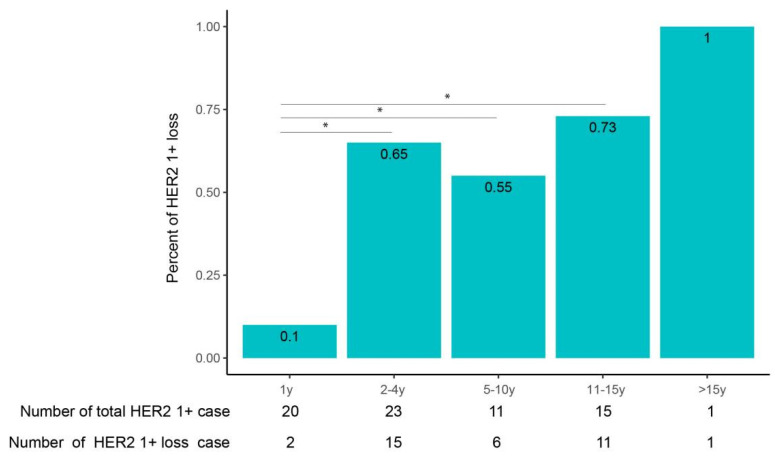
Loss of HER2 intensity in HER2 1+ breast cancer tissue with different stored ages. HER2 1+ breast cancer stored for different ages was selected and restained with HER2. The proportion of cases with loss of HER2 intensity was summarized. * means *p* < 0.05.

**Figure 5 life-14-00031-f005:**
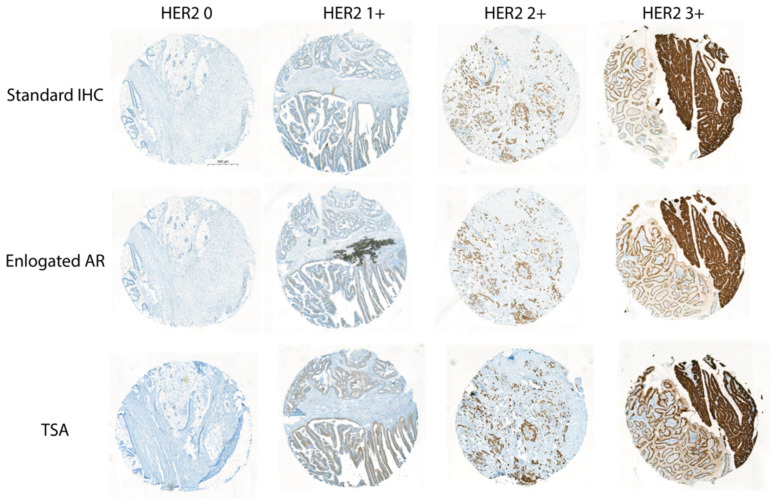
Effect of improved IHC on HER2 staining intensity of control tissue. For both TAR elongating and TSA applying, the control tissues, including HER2 0, 1+, 2+ and 3+, are used to evaluate the intensity change. The image was presented at total magnification of 4×.

**Figure 6 life-14-00031-f006:**
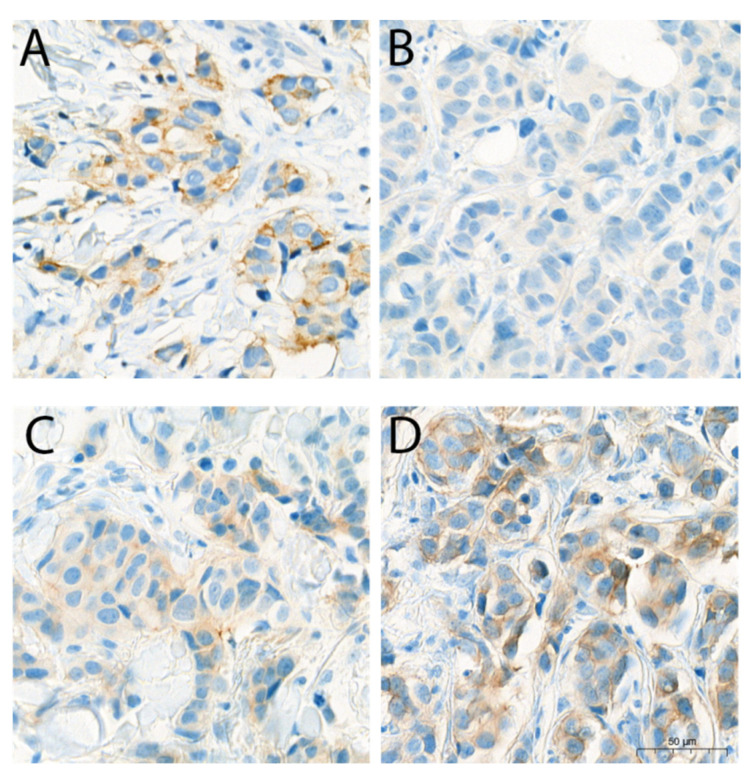
Representative case of HER2 loss from HER2 1+ tissue, which is rescued by both elongating TAR and applying TSA: (**A**) The original HER2 staining image; (**B**) The HER2 staining performed using standard IHC in tissue stored for 5 years; (**C**) HER2 staining intensity after increasing antigen retrieval time; (**D**) HER2 staining intensity after applying TSA. The image was presented at total magnification of 40×.

## Data Availability

The data presented in this study are available in published article.

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
