# Peer review of "Loss of Human Epidermal Receptor 2 Expression in Formalin-Fixed Paraffin-Embedded Breast Cancer Samples and the Rescuing Effect of Enhanced Antigen Retrieval and Signal Amplification"

_life, 2023, doi:10.3390/life14010031_

Round 1

Reviewer 1 Report

Comments and Suggestions for Authors

The topic of the present study is quite interesting and extremely relevant, however  some points need to be clarified.

1. please clearly specify the aim of the study: does it have two aims? the loss of her 2 antigenicity in stored blocks and the rescue of the antigenicity modifying the protocol of staining, maybe?

2. please specify which was the HER2 clone used in FFPE tissue in different period of time

3. the first part of Results should be inserted in Method section

4. To better explain the methods, a flow chart of the study should be added

5. What does  "HER2 loss ratio" mean (line 190)

6. What does AR mean  (line 82)?

Comments on the Quality of English Language

English should be revised

Reviewer 2 Report

Comments and Suggestions for Authors

This manuscript reports on an interesting study on loss of HER2 protein expression in breast cancer tissue archived in paraffin blocks for a longer time. Thereafter, evidence for retrieval of the staining with specific laboratory manipulations is presented. Such observations are already described in the literature, but this study is more detailed especially the retrieval part. The text is well written, the illustrations are of acceptable quality, the references are up-to date. Some suggestions follows:

Main concern: The Discussion is too short. The results shoul be compared to those in the literature. The weaknesses of the study should be listed (small number of cases, no data on pre-paraffin conditions like cold ischeamia, no data on the thickness of the tissue in the paraffin, etc).

Minor problems:

- The title is too complicated. Just a suggestion: Loss of HER2 expression in FFPE breast cancer samples and the rescuing effect of enhanced antigen retrievel. 

- Fulltext name is needed for FFPE at the first appearance in the text (line 46)

. "ratio" is probably not fully correct, "proportion" may be better (line 53)

- "AR" as abbreviation for antigen retrieval is somewhat disturbing as it may be understood as androgen receptors (much more common abbreviation). Use TAR or skip abbreviation.

- Text within the figures Number of of...

Reviewer 3 Report

Comments and Suggestions for Authors

1. FFPE samples should of course be stored in a way that maintain and stabilise the antigenecity of the detected proteins as the HER-2 receptor to allow a later quality assurance or reevaluation of the primary diagnostic result concerning HER-2 status by IHC , from 0 over 1+,2+ to 3+. The authors show that this is not the case, and they document that the destruction can to a degree be diminished by the suggested procedure. But not quite. And therefore the result of a reevaluation will not increase the number of patients with low HER status eligible for the new drugs

Will the authors comment on that ?

So why should it be done ? Is it not too late for these patients possibly misclassified ?

2. The authors seem to believe that a situation with the IHC result 0 shows that the cells do not have any HER-2 receptors. To my knowledge all cells have HER receptors, probably from 5,000 to 50,000 , and in HER-2  2+ and 3+ with or without amplification  up to 50 million.  So the  result 0 simply may mean that IHC is not sensitive enough to measure number of receptors within the normal range . And therefore neither to detect 1+ or 2+ status. Will the authors comment ?

3. The problem with IHC is probably that the formalin fixation destroy to a considerable degree the 3D and antigenic structure of the protein and no reparation proces will be able to correct that. Authors comment ?

Further the cutting of the cells in the tissue leaves only the brim of the cells so most of the evenly distributed surface receptors are not detectable . To preserve the receptor number fresh frozen tissue should be preferred, and quantitative proteomic methods applied for enumeration ( as eg in Olsen et al :Her-2 concentrations in breast cancer cells.Clin Chem Lab Med 2007;45(2):177-182)

4.  Do the authors consider it worthwhile to improve this IHC semiquantitative , insensitive and irreproducable method in the light of the need for better proteomic methodology in precision and personal medicine ?

4. 

Comments on the Quality of English Language

The differences in chinese and western arguing and thinking make some sentences difficult to understand
